# Rationalizing the design of a broad coverage *Shigella* vaccine based on evaluation of immunological cross-reactivity among *S. flexneri* serotypes

**Francesco Citiulo[1] \*, Francesca Necchi[1], Francesca Mancini[1], Omar Rossi[1], Maria Grazia Aruta[1], Gianmarco Gasperini[1], Renzo Alfini[1], Simona Rondini[2], Francesca Micoli[1], Rino Rappuoli[2], Allan Saul[1‡], Laura B. Martin[1‡]**

**1** GSK Vaccines Institute for Global Health S.r.l. (GVGH), Siena, Italy, **2** GSK, Siena, Italy

‡ Senior authors contributed equally to this work.
\* francesco.x.citiulo@gsk.com

## Abstract

No vaccine to protect against an estimated 238,000 shigellosis deaths per year is widely available. *S. sonnei* is the most prevalent *Shigella*, and multiple serotypes of *S. flexneri*, which change regionally and globally, also cause significant disease. The leading *Shigella* vaccine strategies are based on the delivery of serotype specific O-antigens. A strategy to minimize the complexity of a broadly-protective *Shigella* vaccine is to combine components from *S. sonnei* with *S. flexneri* serotypes that induce antibodies with maximum cross-reactivity between different serotypes. We used the GMMA-technology to immunize animal models and generate antisera against 14 *S. flexneri* subtypes from 8 different serotypes that were tested for binding to and bactericidal activity against a panel of 11 *S. flexneri* bacteria lines. Some immunogens induced broadly cross-reactive antibodies that interacted with most of the *S. flexneri* in the panel, while others induced antibodies with narrower specificity. Most cross-reactivity could not be assigned to modifications of the O-antigen, by glucose, acetate or phosphoethanolamine, common to several of the *S. flexneri* serotypes. This allowed us to revisit the current dogma of cross-reactivity among *S. flexneri* serotypes suggesting that a broadly protective vaccine is feasible with limited number of appropriately selected components. Thus, we rationally designed a 4-component vaccine selecting GMMA from *S. sonnei* and *S. flexneri* 1b, 2a and 3a. The resulting formulation was broadly cross-reactive in mice and rabbits, inducing antibodies that killed all *S. flexneri* serotypes tested. This study provides the framework for a broadly-protective *Shigella* vaccine which needs to be verified in human trials.

## Author summary

A strategy to optimize the composition for a broadly-protective *Shigella* vaccine is to combine components directed against *S. sonnei* with *S. flexneri* serotypes to induce antibody

**Data Availability Statement:** All relevant data are within the manuscript and its Supporting Information files.

**Funding:** This work was sponsored by GlaxoSmithKline Biologicals SA which was involved in all stages of the study conduct and analysis.

**Competing interests:** We have read the journal's policy and the authors of this manuscript have the following competing interests: The authors are employees of the GSK group of companies and SR, RR, AS and LBM hold shares in the GSK group of companies. AS was an employee of the GSK group of companies at the time of the study and is now an honorary member of the Burnett Institute, Melbourne, Australia. LBM reports grant from the Bill and Melinda Gates Foundation and Wellcome Trust outside the submitted work. FCI, LBM and AS are inventors of patents owned by the GSK group of companies and relevant to Shigella vaccines.

responses with the maximum cross-reactivity between different serotypes. Based on mouse and rabbit immunogenicity, we selected 4 GMMA-immunogens, derived from *S. sonnei* and *S. flexneri* 1b, 2a and 3a, able to induce antibodies that were broadly bactericidal against most epidemiologically significant *S. flexneri* strains in mice and rabbits. This was not predicted on the basis of O-antigen modifications conferring serotype or group specificities and allowed revisiting the dogma of cross-protection among *S. flexneri* serotypes. Overall, this study provides a framework for the rational design of a broadly-protective vaccine that will be evaluated in upcoming human vaccine trials. It also tackles a key issue regarding *Shigella* vaccine development that is balancing a sufficient number of antigenic components in the vaccine to provide adequate coverage of serotype diversity while minimizing complexity.

## Introduction

*Shigella* infections are endemic throughout the world, but the main disease burden is in developing countries and, particularly in young children. The Global Burden of Disease Study 2017 estimated that *Shigella* causes 15.2% (i.e. 238,000) of the 1.57 million deaths from diarrheal infections [1] with 98.5% of *Shigella* deaths occurring in low- and middle-income countries. Children younger than 5 years of age account for 33% of deaths. Consistent with these global estimates, the prospective Global Enteric Multicenter Study (GEMS) found that shigellosis is one of the top causes of moderate to severe diarrhea (MSD) in children under 5-years-old in 4 sites in sub-Saharan Africa and 3 sites in South Asia [2,3]. Of 1,120 *Shigella* isolates typed, *S. sonnei* was the dominant species (24%) and a range of *S. flexneri* serotypes was found in different sites. Overall, the dominant *S. flexneri* serotype was *S. flexneri* 2, but the distribution of *S. flexneri* serotypes and subtypes varied by location. For example, in the Bangladesh site of the GEMS study the order from the most to the less frequent serotypes was *S. flexneri* 2a, 2b, 3a, 6, 1b, 4a, Y (X was not detected), whereas in Kenya the order was *S. flexneri* 6, 1b, 3a, 4a, 2b, 2a (1a, X, Y were not detected). These data highlight the difficulties in defining the composition of a broadly-protective *Shigella* vaccine for global use, if based solely on the existing epidemiology.

*Shigella* vaccines under development span a spectrum of approaches and antigens [4–6]. Almost all include the O-antigen (OAg) component of the lipopolysaccharide (LPS), which is considered a protective antigen [7], but restricts vaccine efficacy to homologous and cross-reactive serotypes. The OAg of *S. sonnei* has a distinct structure compared to the *S. flexneri* serotype OAg. All serotypes and subtypes of *S. flexneri*, except for serotype 6, share a common OAg backbone (7, 8) with repeating units (RU) of:

$\rightarrow$2)-$\alpha$-L-Rha$p^{\mathrm{III}}$-(1$\rightarrow$2)-$\alpha$-L-Rha$p^{\mathrm{II}}$-(1$\rightarrow$3)- $\alpha$-L-Rha$p^{\mathrm{I}}$-(1$\rightarrow$3)-$\beta$-D-Glc$p$NAc-(1$\rightarrow$.

Modification of the OAg backbone sugars by glucose, acetate or phosphoethanolamine creates the OAg structures specific to the different serotypes (Fig 1B). The enzymes responsible for the OAg backbone modifications are encoded on mobile elements and new *S. flexneri* serotypes and subtypes could emerge by bacteriophage-mediated integration of OAg modification genes [8,9]. The serotypes *S. flexneri* 1, 2, 4, 5 and X are defined by type specificities created by glucosylation. *S. flexneri* 3 is defined by O-acetylation on rhamnose I. *S. flexneri* Y does not contain any of these substitutions and is defined by the absence of serotype specificities. *S. flexneri* 6 has a different OAg backbone composed of two rhamnose, one galacturonic acid and one N-acetylgalactosamine [10,11]:

A

| *S. flexneri* serotype | Type Specificities (MFI) | | | | | | Group Specificities (MFI or PCR) | | | | |
|---|---|---|---|---|---|---|---|---|---|---|---|
| | I | II | III | IV | V | VI | (3)4 | 6 | (7)8 | 9 | 10 |
| 1a | 7500 | 30 | 50 | 60 | 60 | 40 | 7450 | 30 | 40 | + | |
| 1b | 8100 | 50 | 60 | 60 | 50 | 60 | 7200 | 8950 | 30 | + | |
| 1c | 7900 | 60 | 150 | 50 | 60 | 40 | 1550 | 40 | 30 | | |
| 2a | 50 | 9200 | 90 | 60 | 60 | 40 | 5200 | 30 | 25 | + | + |
| 2b | 70 | 8700 | 60 | 60 | 30 | 40 | 30 | 25 | 1530 | | |
| 3a | 30 | 40 | 8500 | 60 | 30 | 60 | 1200 | 6250 | 3540 | | |
| 3b | 50 | 90 | 7200 | 50 | 40 | 80 | 30 | 4500 | 45 | | |
| 4a | 80 | 60 | 30 | 7500 | 30 | 40 | 2350 | 25 | 30 | | |
| 4b | 90 | 80 | 60 | 7600 | 30 | 60 | 20 | 1250 | 30 | | |
| 5a | 30 | 60 | 40 | 80 | 5800 | 80 | 6500 | 40 | 30 | | |
| 5b | 250 | 120 | 170 | 30 | 6600 | 60 | 40 | 30 | 620 | | |
| 6 | 150 | 190 | 120 | 90 | 60 | 2400 | 250 | 30 | 30 | + | |
| X | 80 | 90 | 110 | 60 | 90 | 60 | 20 | 30 | 450 | | |
| Y | 150 | 180 | 90 | 90 | 90 | 80 | 540 | 20 | 20 | | + |

B

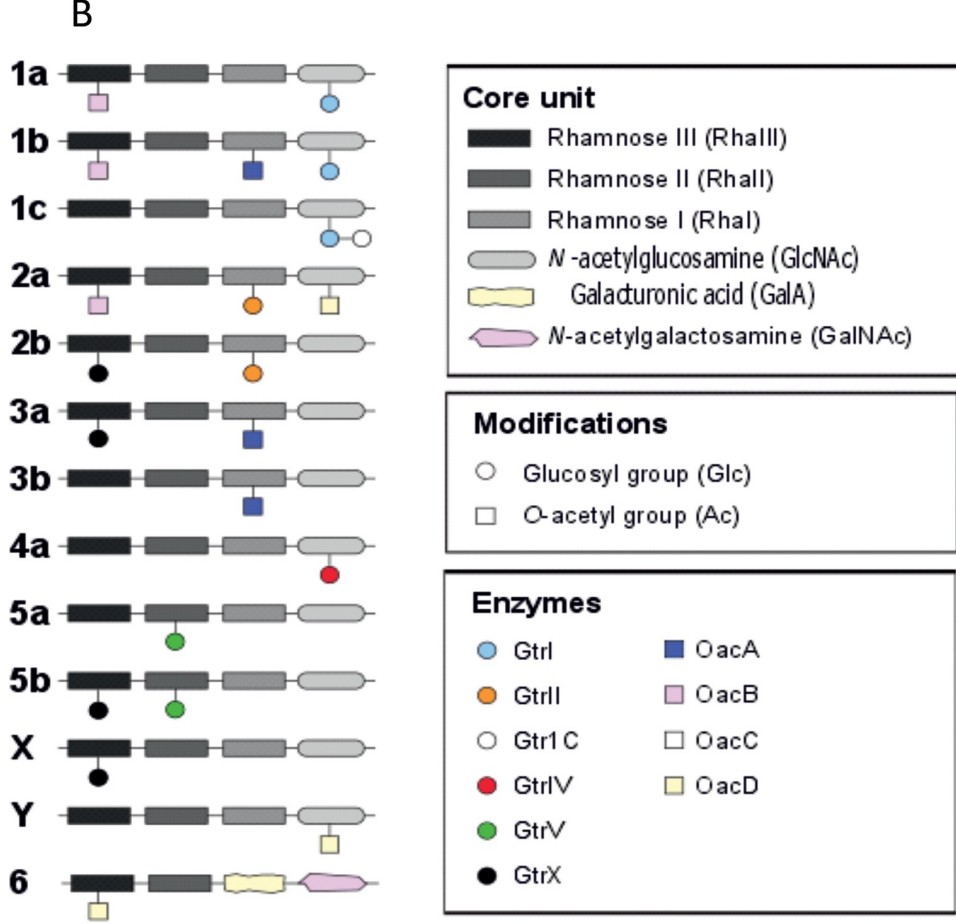

**Fig 1.** Characterization of the *S. flexneri* bacteria used to generate GMMA producing stains in this study based on (A) phenotyping with commercially available type specific antisera or inferred by the presence of genes encoding O-acetylases for groups 9 and 10 specificities as defined by the PCR genotyping (and thus are only typed as positive or negative) and (B) the resulting OAg structure.

→2)-α-L-Rha*p*$^{III}$-(1→2)-α-L-Rha*p*$^{II}$-(1→4)-β-D-Gal*p*A-(1→3)-β-D-Gal*p*NAc-(1→). Although phylogenetically dissimilar (*S. flexneri* 6 is in the *S. boydii* cluster) [12], *S. flexneri* 6 reacts with *S. flexneri* species-specific antisera.

The OAg modifications by glucose, acetate or phosphoethanolamine, alone or in combination, that are common to several of the *S. flexneri* serotypes (including *S. flexneri* serotype 6) generate the group specificities 6, 7(8), 9 and 10. The group specificity 3(4) is related to the unmodified OAg RU. Group specificities are present in various combinations, with at least 27 reported to date [10], and generate shared epitopes that are potential targets of antibodies elicited by cross-protecting vaccines. Group specificity was at the basis of studies by Noriega *et al.* [13], who tested a bivalent vaccine of live attenuated *S. flexneri* 2a and *S. flexneri* 3a in guinea pig conjunctivitis challenge model. Delivery of the bivalent attenuated strains was predicted to cover most *flexneri* strains via the group specificities. In the model, protection against *S. flexneri* serotypes 1b, 2b, 5b and Y was achieved (as predicted based on shared group specificities), but not against serotypes 1a and 4b (contrary to the prediction). Noriega *et al.*, hypothesized that this bivalent vaccine would not have protected against *S. flexneri* 6.

However, subsequent work identified group specificity 9, which is common to all known *S. flexneri* 6 and some (but not all) *S. flexneri* 2a isolates [10]. Thus, protection may have been expected against *S. flexneri* 6 from the bivalent attenuated *S. flexneri* 2a and *S. flexneri* 3a vaccine if it was mediated through group specificities. Moreover, the envisaged coverage of Noriega's bivalent combination in the field would be limited because of the prevalence of *S. flexneri* 1b, 4b and 6, responsible for approximately equal burden of disease, ranking after *S. flexneri* 2a and 3a [14]. To address this, Livio *et al.* proposed including *S. flexneri* 6 in a combination vaccine along with *S. flexneri* 2a and 3a and *S. sonnei*. However, this would still leave people unprotected against *S. flexneri* 1b and 4b. There is a need to determine which *S. flexneri* serotype cross-reactions result in cross-protection and to determine the number and identity of S. flexneri serotypes that must be present in a vaccine to provide acceptable coverage.

In this study, we examined the ability of mouse antisera, raised by immunization with GMMA from 14 subtypes of *S. flexneri*, to bind to *S. sonnei* and a panel of 11 *S. flexneri* subtypes, from all 8 serotypes, by FACS (cross-reaction) and to kill (cross-protection) homologous and heterologous bacterial lines in a complement mediated serum bactericidal assay (SBA). GMMA (Italian *gemma* = bud), are outer membrane exosomes, approximately 50 nm in diameter, released from Gram-negative bacteria genetically modified to induce hyperblebbing [15]. GMMA contain all the outer-membrane components of the parent bacteria and may be engineered to contain a less reactogenic form of LPS expressing the OAg [16]. Thus, GMMA may induce responses similar to those seen with the killed bacterial used in early studies. The GMMA-technology has been used in human trials as *S. sonnei* vaccine [17–20] and is being extended to a multicomponent Shigella vaccine.

## Materials and methods

### Ethics statement

All animal immunization experiments were carried out in compliance with the relevant guidelines of Italy (Italian Legislative Decree n. 116/1992) and the institutional policies of the GSK group of companies. The animal protocol was approved by the Italian Ministry of Health (Approval number AEC 201309).

### *Shigella* strains

*S. sonnei* 53G was obtained from Walter Reed Army Institute of Research, Washington, D.C., USA [21]. The *S. sonnei* ΔvirG::cat strain used in FACS and SBA was generated by Caboni *et al.* [22] to ensure a stable expression of OAg during growth by stabilization of the pSS virulence plasmid that contains the OAg cluster genes by culturing the bacteria in presence of chloramphenicol.

*S. flexneri* lines of the 14 subtypes were purchased from the Public Health England, London, UK. Working cell banks were prepared and typed using both agglutination and surface staining by FACS typing with the commercial *Shigella* typing antisera from Denka Seiken (Shanghai) Co., Ltd; the type specific serum I, II, III, IV, V and grouping sera 3(4), 6 and 7(8). Manufacturer's recommendations were followed for agglutination. For FACS typing, bacteria were grown in LB medium, diluted to $2x10^7$ CFU/mL in PBS, then 50 μL were transferred in 96 well plate on ice, incubated with 1:400 dilution of typing and grouping antisera from Denka Seiken, washed, then incubated with 1:1,000 dilution of fluorescein-conjugated F(ab')2 fragment goat anti-rabbit IgG specific (Jackson Immuno Research Europe Ltd.). The bacteria were then fixed for 3 h with BD Cytofix (containing 4.2% formaldehyde), washed and resuspended in 130 μL PBS. Samples were measured with a BD FACS Canto equipped with a high throughput sample reader using BD FACS DIVA version 8.0.1 software. Bacteria were gated on FSC-A versus SSC-A and the signal measured (FITC/fluorescein channel). Analyses were performed with FlowJo version 10.3 (FlowJo, LLC, Ashland, Oregon). The Mean Fluorescence Intensity (MFI) was used as the measure of staining strength. All bacterial lines gave the expected typing pattern. For *S. flexneri* X the reaction with group 7(8) antisera was weak; this weak reaction was not confirmed in the clone selected for GMMA production. By FACS analysis, an instability of the *S. flexneri* 5b cell line was identified; the population was a mixture of bacteria that were positive or negative for group 7(8) and thus a mixed *S. flexneri* 5a/5b phenotype, presumably due to variable expression of the *gtrX* gene encoding the glycosyl-transferase that distinguishes *S. flexneri* 5a from 5b. This was also true of the *S. flexneri* 5 GMMA producing line and thus the GMMA population used for vaccination were probably a mixture of *S. flexneri* 5a and 5b. For use in the FACS and SBA assays, a new working cell line was selected from the *S. flexneri* 5b bacterial cells that uniformly and strongly reacted with the group 7(8) antisera.

In addition to the serological typing, the bacteria used for the GMMA production and the cell lines of the SBA target panel were genotyped by PCR for the genes encoding the group specificities 9 (*oacB* or *oacC*) and 10 (*oacD*) phenotypes. PCR reaction mixtures contained 12.5 μL DreamTaq Green PCR Master Mix (2x), 9.5 μL sterile water, 1 μL 10 mM forward primer, 1 μL 10 mM reverse primer and 1 μL template (bacteria suspended in water to an OD600 of 5). After amplification, the presence of the amplified gene was detected by electrophoresis using Ethidium bromide stained agarose gels.

### GMMA production, purification and formulation

To generate the GMMA producing lines, the *tolR* gene was deleted as described for the generation of the *S. sonnei* ΔtolR::kan mutant [23]. The resulting clonal lines were re-typed to assure that the genetic modifications had not changed serotype and serogroup specificities. As for the parent line, most, but not all, of the *S. flexneri* 5b GMMA producing bacteria typed by FACS as *S. flexneri* 5a (i.e. negative for group 7(8)).

Bacterial strains were grown at 30˚C on LB agar or in liquid chemically defined medium (SDM), as described [23,24]. When required, kanamycin (30 μg/mL), was added for selection of the GMMA producing strains. For GMMA production, overnight cultures were used to inoculate the SDM at an $OD_{600}$ of 0.03–0.05 and incubated at 30˚C and 200 rpm to an $OD_{600}$ of 8–10.

Culture supernatants were collected by centrifugation followed by a 0.22-μm filtration, ultracentrifuged and the resulting pellet containing GMMA resuspended in PBS as described [24].

GMMA quantities were expressed as total protein using the micro-BCA protein assay (Bio-Rad) kit according to the manufacturer's instructions and bovine serum albumin (Pierce) for the standard curve.

The amount of OAg in the GMMA was determined by HPAEC-PAD analysis by measuring rhamnose content (3 rhamnose residues per OAg RU for all *S. flexneri* serotypes, except *S. flexneri* 6 for which there are 2). The OAg to protein w/w ratio in the GMMA varied from 0.39 to 0.8 (S4 Table). GMMA from *S. flexneri* X contained lower amount of OAg and the OAg /protein ratio was 0.12.

GMMA were adsorbed onto aluminum hydroxide (Alhydrogel 2%, Brenntag Biosector, Denmark). GMMA were added to Alhydrogel to give 4 μg/mL GMMA protein and 0.7 mg $Al^{3+}$/mL in 10 mM Tris, pH 7.4 and 9 g/L NaCl, then stirred for 2h. Formulations were tested to show they had no bacterial contamination and were stored at 2–8˚C for one week prior to use.

## Immunogenicity studies in mice and rabbits

GMMA were used to immunize mice and the resulting sera were included in the cross-reaction panel testing. Animal studies were performed as part of the Italian Ministry of Health Animal Ethics Committee project number 201309. Four CD1 mice per group (female, 4 to 6 weeks old) were immunized intraperitoneally (500 μL each mouse) with 2 μg of GMMA (protein) on days 0 and 21; blood was collected and sera obtained on day 21 and 35 (bleed out). The day 35 sera were pooled and used for the studies reported in this paper. Mice were also immunized with combination of different GMMA maintaining the same GMMA and Alhydrogel content as the individual formulations. New Zealand white Rabbits (3 per group, female > 1.5 kg) were immunized intramuscularly with 500 μL of Sonflex1-2-3 GMMA formulation on days 0 and 28 and sera samples obtained on day 42. The Sonflex1-2-3 vaccine contained the component *S. sonnei* 1790-GMMA at a dose of 25 μg total protein (corresponds to 1.6 μg OAg) and GMMA of *S. flexneri* 1b, 2a and 3a present at 10 μg OAg each; the GMMA were formulated on Alhydrogel (0.7 mg $Al^{3+}$/mL).

## Cross-reactivity of anti-GMMA sera measured by FACS

Prior to assessment of cross-reactivity, all the *S. flexneri* bacteria from the different serotypes used in the study were tested for binding of pooled sera raised against OAg negative *S. flexneri* 2a GMMA using the methodology described below.

Surface staining of the panel of 11 OAg positive *S. flexneri* subtype strains was carried out with the pooled day 35 sera from the 14 immunization groups. As additional control for the study sera were also tested on OAg positive and negative *S. sonnei* and sera raised against OAg negative *S. flexneri* 2a were also tested on OAg negative *S. flexneri* 2a bacteria.

The pooled day 35 sera, were added to the bacterial suspensions in exponential growth phase, thoroughly washed to remove debris, diluted to $2x10^7$ CFU/mL in PBS, and incubated for 1 h, washed again, then APC-conjugated anti-mouse IgG (1:400 dilution) was added and incubated for an additional 1 h. Cells were fixed with BD Cytofix, washed and resuspended in PBS. The signal was measured in the APC channel. The baseline was set by *S. flexneri* 1b, 2a, 3a and 6 controls incubated only with the secondary antibody and without any mouse serum. The results of one of two comparable technical replicates is shown as a matrix with the mean fluorescence intensities (MFI) of surface staining of *S. flexneri* wild type bacteria of the different serotypes (S2 Table). The Log MFI was used to generate the heat map of the binding of *S. flexneri* GMMA antisera on *Shigella* bacterial strains.

## Cross-functionality of anti-GMMA sera measured by high through-put Luminescence—Serum bactericidal assay

SBA for all *S. flexneri* serotypes were developed and performed as described [25]. For each serotype the amount of complement was set based on their inherent sensitivity to Baby Rabbit Complement (BRC) and this highest % of BRC that lacked intrinsic bactericidal activity was selected for subsequent testing. Briefly, *S. sonnei* and *S. flexneri* bacteria derived from the FACS working cell banks were grown to log-phase (OD$_{600}$ of 0.2), diluted 1:1000 in PBS and distributed in 96-well plates. To each well, dilutions of heat-inactivated (30 min at 56˚C) pooled mouse sera and active BRC (7–20% of the final volume) was added. As control, bacteria were incubated with sera plus heat-inactivated BRC, sera alone (no BRC), SBA buffer or active BRC. After 3h incubation, surviving bacteria were determined by measuring ATP. SBA is reported in serum titers, defined as serum dilutions giving 50% inhibition of the ATP level in the positive control (IC50). Titers below the minimum measurable titer of 100 were assigned titer of 10. A matrix showing serum titers on *S. flexneri* wild type cell lines of the different serotypes is reported in S3 Table. The Log IC50 was used to generate the heat map of the bactericidal activity of *S. flexneri* GMMA antisera on *Shigella* bacterial strains.

## Theoretical SBA heat map modelled on typing sera specificities

A theoretical matrix of cross reactivity mediated by specificities serotype (TY) or group 3(4), 6, 7(8), 9 and 10 was generated showing the specific shared specificities among the immunizing GMMA and the target bacterial strain. Then, we used the observed Log IC50 of sera tested on the homologous and heterologous serotypes to assign an hypothetical strength of cross-functionality; the observed average Log IC50 for sera tested on the homologous serotypes (i.e. anti-*S. flexneri* 2a antisera tested on *S. flexneri* 2a or on *S. flexneri* 2b) was 4.7. Therefore, in constructing a theoretical SBA heat map, the SBA Log IC50 for sera tested on homologous serotypes was assigned a value of 4.7. The observed average SBA Log IC50 tested on heterologous serotypes sharing a single common typing specificity, where the SBA was measurable, was 3.9. Thus, where an immunizing GMMA shared a single strongly typing group specificity, we assigned a value of 3.9 to this interaction. As shown in S1 Table typing of the target bacteria with the standard group-specific reagents showed several strains that gave positive but weak interaction with typing reagents. On average these had Log MFI that were 0.9 Log units lower for group 3(4) or 0.7 for group 7(8) than the high responders. In this case, we assigned a value of 3.1 (i.e. 0.8 Log units lower than the high responders) to the modelled SBA value (we assumed that a weakly typing positive GMMA producing strain still had sufficient group specific antigen to elicit a full group specific antibody response). Where the immunizing GMMA and the target bacteria shared two group specificities, we assigned a Log IC50 titer as the Log of the sum of the titers. Thus, the modelled titer of anti-*S. flexneri* 1a GMMA on *S. flexneri* 2a that share both the group 3(4) and the group 9 specificities was assigned a Log IC50 of $4.2 = Log (10^{3.9} + 10^{3.9})$. For the modelled SBA titers, a calculated Log IC50 that could be obtained by immunizing with a mixture of *S. flexneri* 1a and 3a was calculated similarly; e.g. the estimated Log IC50 of a mixture of anti-*S. flexneri* 1b and 3a GMMA on *S. flexneri* 2a was $4.0 = Log (10^{3.9} + 10^{3.1})$.

## Results

### Serotype and group specificities of the *S. flexneri* strains used in this study

A characterization of the bacteria used in this study, based on typing with commercially available specific antisera or inferred by the presence of genes encoding O-acetylases, are shown in

Fig 1A and the corresponding OAg structures are shown in Fig 1B. The presence of O-acetylation on the OAg was demonstrated by NMR for *S. flexneri* 1b, 2a and 3a. Details of the typing are included in the Supplementary Information (S1 Table).

## Evaluation of cross-reactivity by FACS of antisera raised in mice against GMMA from one subtype of *S. flexneri* on homologous and heterologous *S. flexneri* subtypes

To visualize the cross-reactivity patterns, a heat map was generated with the Log10 of the Mean Fluorescence Intensities (Log MFI) of surface staining of a panel *S. flexneri* bacteria (Fig 2A). The detailed MFI values are reported in the Supplementary Information (S2 Table). A threshold criterion was applied to distinguish relevant cross-reactivity, defined by cross-reactivity that could be predictive of field cross-coverage as opposed to low level cross-reactivity unlikely to provide field cross-coverage. The threshold criterion was based on published preclinical animal data where there was an ability to protect against heterologous challenge from *S. flexneri* 2b in guinea pigs immunized with attenuated *S. flexneri* 3a [13]; this threshold is arbitrary in nature, given that it is based on only one heterologous assessment and in a different animal model. In our study for FACS experiments, the relevant threshold criteria was shown to be MFI $\geq$ 130 (Log MFI 2.1), that in our assay is 10-fold above background. By assessing the staining intensity of the antisera raised by the GMMA serotypes used for immunization on homologous and heterologous binding serotypes, it was possible to identify broad specificity immunogens.

To better understand the antibody responses elicited by GMMA immunization and the contribution of the GMMA OAg and protein components, we also tested binding of mouse antisera raised against OAg negative or OAg positive GMMA on OAg positive or OAg negative bacteria.

**Binding of antisera raised against OAg positive GMMA.** All antisera had strong homologous subtype binding, with MFI ranging from 4,508 for *S. flexneri* 5b (Log MFI 3.7) to 98,520 for *S. flexneri* 2b (Log MFI 5.0). The exception was anti-*S. flexneri* X that gave relatively weak binding to *S. flexneri* X bacteria (MFI 541, Log MFI 2.7).

For most of the antisera tested, the highest level of cross-reaction was identified among homologous serotypes (*S. flexneri* serotypes having a common glucosyl or acetyl modification at the same position on the OAg backbone e.g. *S. flexneri* 1c antisera binding to *S. flexneri* 1a and 1b bacteria). The level of heterologous cross-reactivity varied; antisera from *S. flexneri* 2a GMMA reacted strongly only with the homologous serotypes and weakly with only two other serotypes *S. flexneri* 4a and Y (i.e. with an MFI > 130 for 2 of 8 heterologous serotypes tested). By contrast, antisera against *S. flexneri* 1b GMMA elicited broad cross-reactions to homologous serotypes and most heterologous serotypes giving an MFI >130 to 7 of 8 heterologous serotypes. Thus, *S. flexneri* 1b, 1c, 3b, 4a, 5a and 5b GMMA are broad-specificity immunogens by FACS (MFI > 130 on $\geq$ 60% heterologous serotypes/subtypes); *S. flexneri* 1a, 2b, 3a and X, medium-specificity immunogens (MFI > 130 on 50% to < 60% heterologous serotypes/subtypes) and *S. flexneri* 2a, 6 and Y, narrow-specificity immunogens (MFI > 130 on < 50% heterologous serotypes/subtypes). *S. flexneri* 4b GMMA antisera had an indeterminate breadth of specificity because it failed to generate strong binding to homologous serotypes (i.e. *S. flexneri* 4a) and to OAg negative bacteria; the lack of binding to other serotypes may only be indicative of a poor immunogenicity of this GMMA.

Bacterial subtypes varied considerably in their ability to be recognized by heterologous GMMA antisera. Some of the *S. flexneri* subtypes were widely recognized by many different antisera, specifically *S. flexneri* 1a, 4a, 5b, 6, X and Y. Thus, these bacteria are broad-specificity

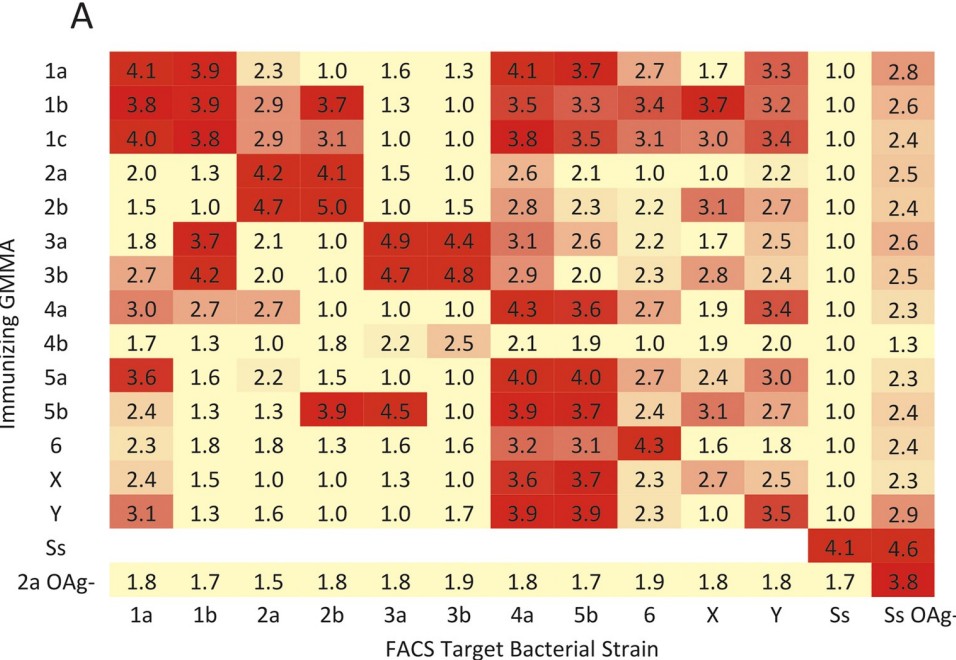

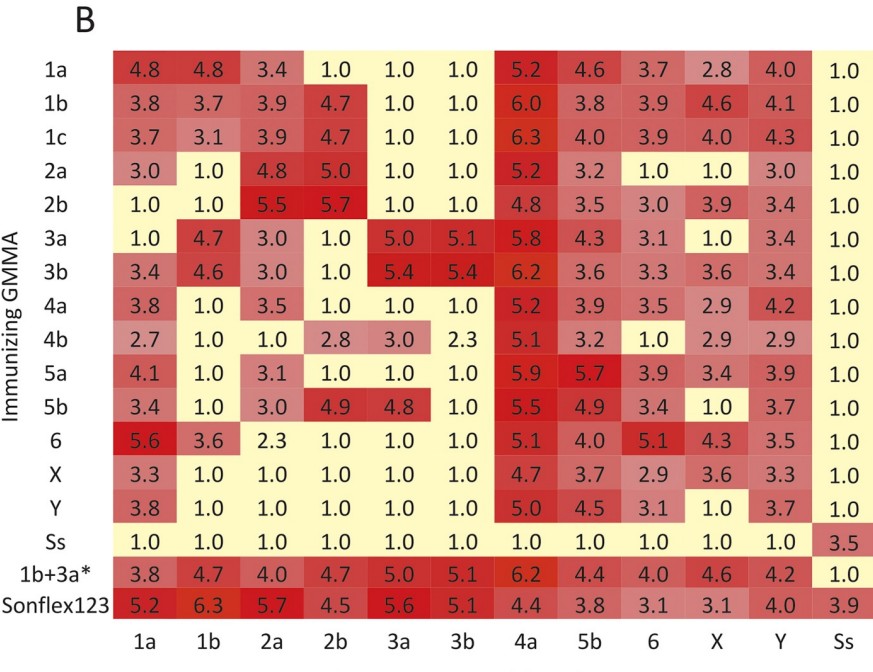

**Fig 2.** Experimentally derived heat maps of the interaction of *S. flexneri* GMMA mouse antisera with *Shigella* bacterial strains assessed by (A) binding measured by FACS Log MFI, threshold Log MFI > 2.1; (B) functional activity as measured by Serum Bactericidal Activity Log (titer) of single components and the Sonflex1-2-3 (Sonflex) formulation eliciting broad coverage, threshold Log IC50 ≥ 2.7. *Calculated SBA titers predicted by immunizing with a *S. flexneri* 1b and 3a mixture based on cross-reactions observed from mice immunized with the individual GMMA. Ss: *Shigella sonnei*, all other GMMA and bacteria are *S. flexneri* serotypes; OAg-: bacterial that lack OAg. Color coding: Light yellow = weak antibody recognition (at or below threshold) to dark red = strong antibody recognition.

targets. By contrast, some subtypes were only recognized by a few antisera. *S. flexneri* 3b was the most restricted target, only recognized strongly by homologous antisera raised against *S. flexneri* 3a or 3b GMMA and weakly by antisera raised against *S. flexneri* 4b GMMA. *S. flexneri* 3a was the next most restricted target recognized only by *S. flexneri* 3b and 5b GMMA antisera. By these criteria, *S. flexneri* 1b, 2a, 2b, 3a and 3b are narrow-specificity targets. As expected, *S. sonnei* bacteria were not stained by any of the *S. flexneri* GMMA antisera.

Binding to OAg negative *S. sonnei*: All antisera raised with OAg positive GMMA gave detectable binding to OAg negative *S. sonnei* (phase II strain). Anti-*S. flexneri* 4b had the weakest binding (MFI 40). All antisera including anti-*S.* flexneri 4b, gave an MFI that was more intense on the OAg negative *S. sonnei* than to at least one of the heterologous OAg positive *S. flexneri* strains tested.

**Binding of sera raised against OAg negative GMMA.**   Antisera raised against OAg negative *S. flexneri* 2a GMMA (GMMA from *S. flexneri* 2a Δ*tolR* Δ*rfbG*) gave strong fluorescence signal on both OAg negative *S. flexneri* 2a (MFI 5,000) and OAg negative *S. sonnei* (MFI 6,300) bacteria; binding of OAg negative *S. flexneri* 2a GMMA antisera was weak on all OAg positive bacteria tested, including OAg positive *S. flexneri* 2a (Fig 2A). These results were in agreement with the work of Mancini *et al*. 2021, that demonstrated that OAg chains shield bacteria from anti-protein antibodies [26], and confirm that anti-OAg antibodies from mice immunized with OAg positive GMMA were the main drivers of bactericidal activity against OAg-positive bacteria.

## Evaluation of cross-functionality by Serum Bactericidal Activity (SBA) of mouse antisera raised against GMMA from one subtype of *S. flexneri* on homologous and heterologous *S. flexneri* subtypes

To visualize the cross-functionality patterns, a heat map of SBA data containing the Log IC50 of pooled antisera on *S. flexneri* bacterial cell lines is shown in Fig 2B. The detailed IC50 titers are reported in the supplementary information (S3 Table). Similar to the FACS data, a threshold criterion was applied to distinguish relevant cross-protection. This threshold criterion was estimated as an IC50 $\geq$ 500 (Log IC50 2.7), that in our assay is 5-fold higher the minimum measurable titer of 100. By assessing the antisera killing capabilities of the vaccinating GMMA serotypes, it was possible to further identify broad-specificity immunogens.

All antisera gave strong homologous bactericidal activity (IC50 > 1000), with $IC_{50}$ ranging from 4,294 for *S. flexneri* X (Log $IC_{50}$ 3.6) to 511,451 for *S. flexneri* 2b (Log $IC_{50}$ 5.7).

For most of the antisera tested, the highest level of cross-killing was identified among homologous serotypes. Antisera from *S. flexneri* 2a GMMA strongly killed homologous serotypes and only weakly the serotypes *S. flexneri* 1a, 4a, 5b and Y. By contrast, antisera against *S. flexneri* 1b GMMA elicited broad cross-killing to most of the heterologous serotypes; corroborating cross-reactivity as identified by FACS.

## Concordance of FACS and SBA data

In fact, the antisera cross-reactivity as judged by FACS (binding) and by SBA (functionality) were similar (Fig 2A compared to 2B) with few inconsistencies. Antisera against *S. flexneri* 4a and 5b GMMA gave moderate intensity MFI on *S. flexneri* 1b but failed to give detectable SBA titers. *S. flexneri* 3b, 4b and 6 GMMA antisera gave relatively stronger SBA titers on more targets than expected from the FACS data. The breadth of the specificity was a little greater as judged by SBA than FACS. Thus, the *S. flexneri* 1b, 1c, 3b, 5a and 5b serotypes GMMA, identified as broad-specificity immunogens by FACS, were joined by *S. flexneri* 3a and 6 (IC50 $\geq$ 1,000 on $\geq$ 60% heterologous serotypes). By SBA, there were fewer medium

specificity immunogens (IC50 ≥ 1,000 on 50% to < 60% on heterologous serotypes, *S. flexneri* 1a and 2b), while *S. flexneri* 2a joined *S. flexneri* 4b, X, Y as narrow-specificity immunogens (IC50 ≥ 1000 on < 50% heterologous serotypes).

Alternatively, the heat map of cross-functionality (SBA data) poorly correlated with a heat map of the theoretical cross-reactivity generated from the expected interactions of serotype and group specific typing antibodies (Fig 3A and 3B). This is contrary to what has been generally assumed that heterologous cross-reactivity (eg, not sharing the same type specificity) would be mediated by the group specificities (ie, 3(4), 6, 7(8), 9 and 10); that is the assumption at the basis of the theoretical matrix Fig 3A, used to generate the theoretical heat map Fig 3B. For example, antisera raised against *S. flexneri* 1b and 1c GMMA gave some of the strongest binding observed to *S. flexneri* 2b, that share no known group specificities. Also present are examples of expected cross-reaction when none was observed experimentally (e.g., antisera raised against *S. flexneri* 2b GMMA on *S. flexneri* 3a target bacteria).

## Generation of a multicomponent *Shigella* vaccine with the identified broad-specificity immunogens

Based on the *S. flexneri* cross-reactivity data, we rationally designed the composition of a vaccine for broad coverage. Immunogenicity in mice revealed that a minimal two-component vaccine including *S. flexneri* 1b and 3a GMMA can provide coverage against virtually all *S. flexneri* serotypes (Fig 2A and 2B). There are reports that humans recognize some *Shigella* serospecificities differently to mice [27] and thus, we decided to include *S. flexneri* 2a GMMA because of its epidemiological relevance. *S. sonnei* was also included due to its global disease burden.

We immunized mice with a 4-component vaccine containing GMMA from *S. sonnei* and *S. flexneri* 1b, 2a and 3a formulated on Alhydrogel, that we named Sonflex1-2-3, and tested the resulting sera for SBA on the panel of 11 *S. flexneri* subtypes from all 8 serotypes and *S. sonnei* (Fig 2B). As predicted, Sonflex1-2-3 antisera showed high cross-functionality, successfully killing the 11 *S. flexneri* serotypes tested as well as *S. sonnei*. Similar experiments were performed in rabbits to evaluate functional activity of antisera raised against single GMMA components and Sonflex1-2-3 demonstrating induction of cross-functional antisera able to kill all the 8 *S. flexneri* subtypes and *S. sonnei* (Fig 4).

## Discussion

A broadly-protective vaccine against shigellosis needs to cover *S. sonnei* and multiple *S. flexneri* serotypes. A challenge to design a practical vaccine is balancing coverage versus cost. There are only a few reports of immunological cross-reactivity among *S. flexneri* serotypes and subtypes in the literature [13,28] and an extensive immunologic screening using preclinical animal models to identify cross-reactive antibodies has not been carried out. Taking advantage of the simplicity of GMMA-technology, we report the generation of a variety of *S. flexneri* GMMA serotypes, their immunogenicity in mice and rabbits and the resulting cross-reactivity and cross-functionality of the antisera.

We used FACS and SBA, two techniques that give a direct measure of the interaction between host antibody response and infective bacteria. As GMMA contain the majority of the outer membrane components of their parent bacteria [16], they can elicit antibodies that bind to antigens present on the bacterial surface, predominantly OAg but also proteins. Indeed, as measured by FACS, OAg negative GMMA (i.e. *S. flexneri* 2a Δ*tolR* Δ*rfbG* GMMA) elicit antibodies that strongly bind to bacteria without OAg, suggesting that the GMMA can induce a broad range of antibody responses. However, some observations from this study show that the

**A**

| Immunizing GMMA | 1a | 1b | 2a | 2b | 3a | 3b | 4a | 5b | 6 | X | Y |
|---|---|---|---|---|---|---|---|---|---|---|---|
| 1a | TY | TY | 3(4) 9 | | 3(4) | | 3(4) | | 9 | | 3(4) |
| 1b | TY | TY | 3(4) 9 | | 3(4) 6 | 6 | 3(4) | | 9 | | 3(4) |
| 1c | TY | TY | 3(4) | | 3(4) | | 3(4) | | 3(4) | | 3(4) |
| 2a | 3(4) 9 | 3(4) | TY | TY | 3(4) | | 3(4) | | 9 | | 3(4) 10 |
| 2b | | | TY | TY | 7(8) | | | 7(8) | | 7(8) | |
| 3a | 3(4) | 6 | 3(4) | 7(8) | TY | TY | | 7(8) | 3(4) | 7(8) | 3(4) |
| 3b | | 6 | | | TY | TY | | | | | |
| 4a | 3(4) | 3(4) | 3(4) | | 3(4) | | TY | | 3(4) | | 3(4) |
| 4b | | 6 | | | 3(4) | 6 | TY | | | | |
| 5a | 3(4) | 3(4) | 3(4) | | 3(4) | | 3(4) | TY | 3(4) | | 3(4) |
| 5b | | | | 7(8) | 7(8) | | | TY | | 7(8) | |
| 6 | 3(4) 9 | 9 | 3(4) 9 | | 3(4) | | 3(4) | | TY | | 3(4) |
| X | | | | 7(8) | 7(8) | | | 7(8) | | TY | |
| Y | 3(4) | 3(4) | 3(4) 10 | | 3(4) | | 3(4) | | 3(4) | | TY |

SBA Target Bacterial strain

**B**

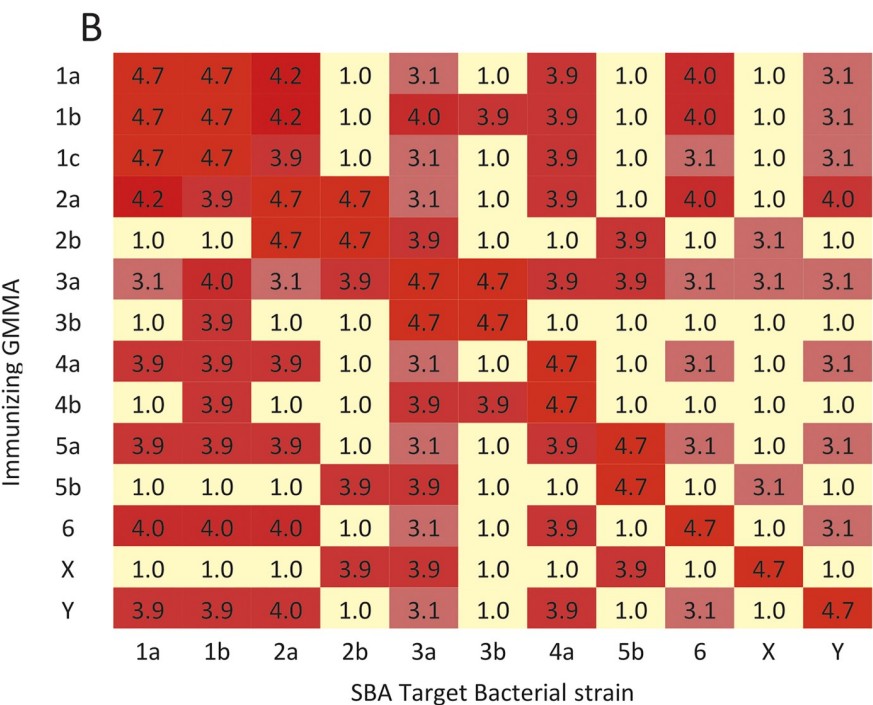

| Immunizing GMMA | 1a | 1b | 2a | 2b | 3a | 3b | 4a | 5b | 6 | X | Y |
|---|---|---|---|---|---|---|---|---|---|---|---|
| 1a | 4.7 | 4.7 | 4.2 | 1.0 | 3.1 | 1.0 | 3.9 | 1.0 | 4.0 | 1.0 | 3.1 |
| 1b | 4.7 | 4.7 | 4.2 | 1.0 | 4.0 | 3.9 | 3.9 | 1.0 | 4.0 | 1.0 | 3.1 |
| 1c | 4.7 | 4.7 | 3.9 | 1.0 | 3.1 | 1.0 | 3.9 | 1.0 | 3.1 | 1.0 | 3.1 |
| 2a | 4.2 | 3.9 | 4.7 | 4.7 | 3.1 | 1.0 | 3.9 | 1.0 | 4.0 | 1.0 | 4.0 |
| 2b | 1.0 | 1.0 | 4.7 | 4.7 | 3.9 | 1.0 | 1.0 | 3.9 | 1.0 | 3.1 | 1.0 |
| 3a | 3.1 | 4.0 | 3.1 | 3.9 | 4.7 | 4.7 | 3.9 | 3.9 | 3.1 | 3.1 | 3.1 |
| 3b | 1.0 | 3.9 | 1.0 | 1.0 | 4.7 | 4.7 | 1.0 | 1.0 | 1.0 | 1.0 | 1.0 |
| 4a | 3.9 | 3.9 | 3.9 | 1.0 | 3.1 | 1.0 | 4.7 | 1.0 | 3.1 | 1.0 | 3.1 |
| 4b | 1.0 | 3.9 | 1.0 | 1.0 | 3.9 | 3.9 | 4.7 | 1.0 | 1.0 | 1.0 | 1.0 |
| 5a | 3.9 | 3.9 | 3.9 | 1.0 | 3.1 | 1.0 | 3.9 | 4.7 | 3.1 | 1.0 | 3.1 |
| 5b | 1.0 | 1.0 | 1.0 | 3.9 | 3.9 | 1.0 | 1.0 | 4.7 | 1.0 | 3.1 | 1.0 |
| 6 | 4.0 | 4.0 | 4.0 | 1.0 | 3.1 | 1.0 | 3.9 | 1.0 | 4.7 | 1.0 | 3.1 |
| X | 1.0 | 1.0 | 1.0 | 3.9 | 3.9 | 1.0 | 1.0 | 3.9 | 1.0 | 4.7 | 1.0 |
| Y | 3.9 | 3.9 | 4.0 | 1.0 | 3.1 | 1.0 | 3.9 | 1.0 | 3.1 | 1.0 | 4.7 |

SBA Target Bacterial strain

**Fig 3. Theoretical cross-reactivity generated from the expected interactions from shared serotype and group specificities.** (A) Theoretical matrix of cross-reactivity mediated by specificities (ie, serotype (TY) or group 3(4), 6, 7 (8), 9 and 10). The specific shared group specificities among the immunizing GMMA and the target bacterial strain that would dictate the cross-reactivity are shown. (B) Theoretical heat map of predicted Log IC50 based on shared serotype and group specificities. All GMMA and bacteria are *S. flexneri* serotypes; Color coding: Light yellow = weak antibody recognition to dark red = strong antibody recognition. See Materials and Methods section for explanation of mathematical derivation of IC50 values.

antisera induced by OAg positive GMMA evaluated by FACS and SBA on OAg positive bacteria are predominantly directed against the OAg:

1. Both FACS and SBA responses were predominantly serotype or subtype specific and no *S. flexneri* GMMA induced antibodies that interacted with *S. sonnei*, which has a different OAg structure but similar LPS core oligosaccharide [29] and outer membrane protein composition [16].

2. Antisera raised against OAg negative GMMA had very weak detectable binding to OAg positive bacteria but very strong binding to OAg negative *S. sonnei* (or *S. flexneri* 2a) bacteria. Conversely, sera raised against OAg positive *S. flexneri* or *S. sonnei* GMMA bound to OAg negative bacteria.

Our finding that antisera generated against OAg negative GMMA were unable to bind to OAg positive bacteria (S1 Fig) agrees with earlier immunization studies using intact bacteria [30]. As well, these results agreed with the recent work of Mancini *et al.* 2021, demonstrating that anti-protein antibodies are induced in mice upon immunization with either OAg-negative or OAg-positive GMMA but the presence of OAg chains on the bacteria surface prevent anti-protein antibody binding and anti-protein mediated bactericidal activity [26]. Both studies suggest that the OAg shields the bacteria from antibody directed against outer membrane antigens and are consistent with the observation that immunity in humans elicited by attenuated *Shigella* strains is predominantly OAg specific [31–33]. This finding is reinforced by the functional analysis of human sera from a Phase 2b study of a *Shigella sonnei* GMMA-based vaccine [20,34]. The adsorption of anti-OAg antibodies from post-vaccination sera confirmed that anti-protein antibodies are not able to induce complement mediated bactericidal killing against *S. sonnei* OAg-positive bacteria [20].

Thus, the anti-OAg specificities induced by GMMA immunization will be important for inducing broad protection. However, given the complex mechanism by which *Shigella* invades the intestinal lumen and establishes infection, this does not rule out protection via other immune mechanisms not involving OAg antibodies, e.g. cell mediated responses against intracellular *Shigella* [35–38].

*Shigella* antisera cross-reactivity has been generally assumed to be mediated by the group specificities (i.e. epitopes 3(4), 6, 7(8), 9 and 10) and this was the basis of the experimental cross-protective vaccine developed by Noriega *et al.* [13]. Instead, we found that the pattern of cross-reactivity observed with the larger panel of GMMA antisera was unexpected in that it could not be assigned to *S. flexneri* serogrouping. Additionally, the theoretical cross-protection, modelled on shared group typing antibody specificities, had little resemblance to what was observed by SBA (Fig 3B compared to Fig 2B). Similar outcomes were obtained by simply scoring the cross-reaction as detectable or not detectable on the basis of the theoretical matrix in Fig 3A, without the detailed assumptions used to create Fig 3B.

Therefore, we conclude that most of the GMMA antisera cross-reactivity cannot be explained by the defined group-specificities derived from highly adsorbed typing sera [39–41]. Presumably, there are serotype-common epitopes within the OAg RU of the different *S.*

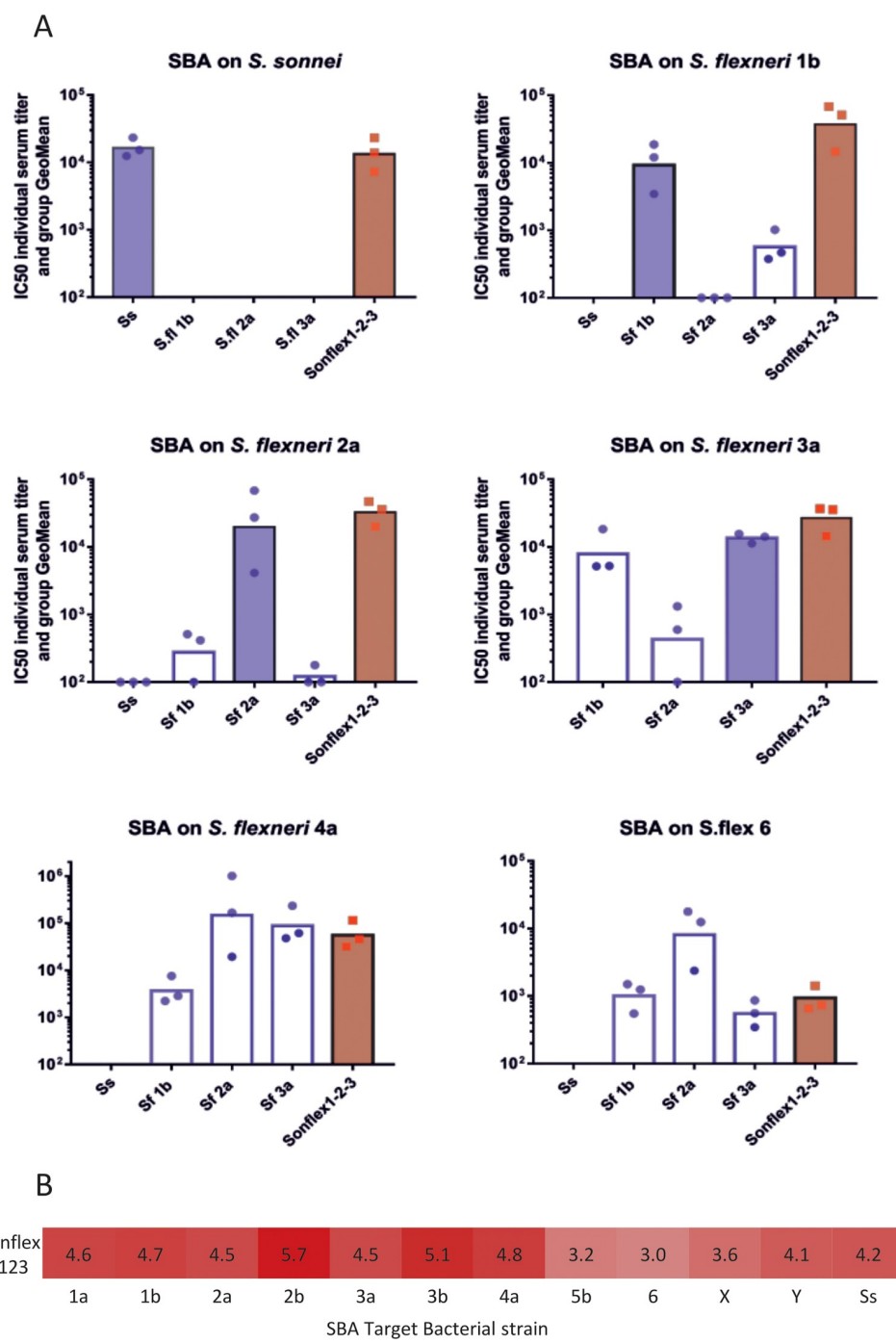

**Fig 4. Functional activity as measured by Serum Bactericidal Activity of sera raised in rabbits against single components and the Sonflex1-2-3 (Sonflex).** (A) SBA titers reported as IC50 of sera from rabbits immunized with Alhydrogel formulated *S. sonnei* (Ss), *S. flexneri* 1b (Sf1b), *S. flexneri* 2a (Sf2a) or *S. flexneri* 3a (Sf3a) GMMA alone or when combined into Sonflex1-2-3. SBA titers against homologous bacterial serotype targets or *S. flexneri* serotypes 4a and 6 are shown. Bars are the average IC50 of 3 rabbits; individual rabbit titers are shown by scatter plot. SBA are presented for *Shigella* bacteria and groups of animals immunized with individual homologous GMMA (blue bars), heterologous GMMA (open bars) or the combination of GMMA (Sonflex1-2-3) containing the homologous GMMA as one of the 4 components (red bar). (B) Heat map of SBA titers, Log (IC50), of the Sonflex1-2-3 formulation antisera tested on different *S. flexneri* serotypes and *S. sonnei* (Ss). Color coding is the same as defined for Fig 2B: Light yellow = weak antibody recognition (at or below threshold) to dark red = strong antibody recognition. Thresholds for shading the heat maps are further detailed in the text.

*flexneri* serotypes that remain to be defined on a molecular basis. This evidence is also supported by the finding that an anti-LPS IgG mAb, F22-4, binds to an epitope formed by two consecutive *S. flexneri* 2a OAg RU [42], rather than an epitope contained within a defined OAg RU.

We found that *S. flexneri* 6 bacteria can be killed by different *S. flexneri* GMMA antisera (i.e., *S. flexneri* 1b) despite being phylogenetically dissimilar [12]. This may be due to antibodies directed against the trisaccharide epitope generated by the junction between adjacent OAg RU: →3)-β- D-Gal$p$NAc-(1→2)- $\alpha$ -L-Rha$p^{III}$-(1→,2)-α-L-Rha$p^{II}$-(1→

Another finding of the study was the lack of reciprocity between immunogen and target. For example, antisera against *S. flexneri* 1b GMMA generated substantial cross-reactivity (by FACS and SBA) against 8 of 10 heterologous OAg positive *S. flexneri* subtype bacterial lines (no cross-reactivity was observed against *S. flexneri* 3a and 3b). In contrast, only antisera raised against *S. flexneri* 3a, 3b and 6 heterologous GMMA showed significant cross-reaction on *S. flexneri* 1b. This effect is possibly caused by OAg RU modifications that either results in partial loss of accessible conformational spaces for their nearest glycosidic linkages [43,44] or generates changes in the spatial conformation of the OAg RU [45] resulting in the inability of antibodies generated against the OAg backbone to bind to the highly modified OAg of *S. flexneri* serotypes (i.e. on the OAg of *S. flexneri* 1b or 3a). On the other hand, many OAg RU modification combinations are not possible in nature due to phage incompatibility [8], result in competitive disadvantage to the invasive capacity of the bacterium by altering the T3SS needle or impair survival to innate immune effectors [46]. Therefore, only few *S. flexneri* serotypes, like *S. flexneri* 1b and 3a, are unique in that they are both broad specificity immunogens and narrow specificity targets and thus, of a high importance for inclusion in a broadly protective multicomponent vaccine.

In fact, the expected cross-reactivity of mouse antisera, against a combined formulation of *S. flexneri* 1b and 3a GMMA, was so broad that it elicited antibodies reacting strongly with all *S. flexneri* isolates tested (Fig 2A and 2B). The opposite, narrow specificity immunogens and board specific targets, was also observed; GMMA from *S. flexneri* 2a, 4b, X and Y, and to a lesser extent *S. flexneri* 6, generated antisera that reacted with relatively few bacterial isolates. Their homologous bacteria were commonly recognized by antisera from other GMMA serotypes suggesting that their inclusion in a vaccine would be less critical due to coverage through cross-reactions with other serotypes.

The lack of correlation between binding and functional cross-reactivity (Fig 2A and 2B) with the classically defined type and group specificities (Fig 3A and 3B) limits the design of combination vaccines. Our observations of extensive cross-reactivity of mouse antisera against broadly specific GMMA immunogens, such as *S. flexneri* 1b and 3a GMMA, is encouraging. It suggests that a simple three component *Shigella* vaccine for GMMA from *S. sonnei*, *S. flexneri* 1b and 3a may be possible and could cover the most of the epidemiologically significant *Shigella* strains, with limited components. However, our data are generated in mice and there is at least one report showing mouse results may not translate to humans [27]. Similar to our *S. flexneri* 2a GMMA results (Fig 2), mice immunized with a *S. flexneri* 2a OAg conjugate did not elicit antibody that reacted with *S. flexneri* 6 OAg, although sera from people immunized with the *S. flexneri* 2a OAg conjugate did elicit antibody that bound to *S. flexneri* 6 OAg and may have contributed to protection of children against infection with *S. flexneri* 6 [27]. Interestingly, we identified that sera from rabbits immunized with *S. flexneri* 2a GMMA elicited SBA against *S. flexneri* 6 similarly to what was observed in humans with the *S. flexneri* 2a OAg conjugate (Fig 4A). Evaluation of antibody cross-reactivity, by ELISA and SBA to the prevalent *S. flexneri* isolates, is planned as part of clinical evaluation of *Shigella* vaccines in humans.

An additional limitation of this study includes lack of any preclinical *in vivo* evaluation of cross-protection. Two preclinical models have been used extensively in the *Shigella* field for vaccine evaluation: mouse pulmonary lung and guinea pig keraconjunctivitis models [47–49]. Although both models rely on mucosal protection, neither are infected via the intestinal tract (route of human exposure). Nor has protection from disease in these preclinical models been directly correlated with human outcomes.

Using the results of the current study, we are advancing development of a 4-component vaccine, containing GMMA from *S. sonnei*, *S. flexneri* 1b, *S. flexneri* 2a and *S. flexneri* 3a formulated on Alhydrogel. We demonstrated that both mouse and rabbit antisera raised against this combination elicited strong SBA against the panel of 11 *S. flexneri* subtypes from all 8 serotypes (Figs 2B and 4B). Based on this data, our combination is likely to have a wider impact on shigellosis, caused by circulating isolates of *S. flexneri* and *S. sonnei*, than the vaccine composition proposed by Noriega and Livo [13,14] containing OAg from *S. sonnei*, *S. flexneri* 2a, *S. flexneri* 3a and *S. flexneri* 6. Interestingly, Barry and Levine have recently suggested that *S. flexneri* 1b could be a useful addition to a multicomponent Shigella vaccine [50]. Clearly, careful analysis of the human sera specificity coming from vaccines trials will be a high priority for demonstrating the vaccine's broad-coverage or for improving the composition, if necessary. Due to the ease of generating and testing GMMA, the approach described here can be applied to the development of many Gram-negative bacterial vaccine candidates to address the challenges of serotype diversity and vaccine coverage, ensuring the inclusion of only the necessary vaccine components, while reducing the complexity of manufacturing.

## Supporting information

**S1 Table. Characterization of *S. flexneri* strains used in the study by slide agglutination and FACS typing.**
(DOCX)

**S2 Table. Experimentally derived matrix of surface staining of *S. flexneri* serotypes reported as Mean Fluorescence Intensities.**
(DOCX)

**S3 Table. Experimentally derived Serum Bactericidal Activity (SBA) Titers reported at IC50.**
(DOCX)

**S4 Table. OAg to Protein ratios and OAg population of *S. flexneri* GMMA from the different serotypes.**
(DOCX)

**S1 Fig. Antisera generated against OAg negative GMMA were unable to bind to OAg positive bacteria.**
(DOCX)

## Acknowledgments

We thank Christiane E. Gerke for important discussion especially in the early planning of a broadly-protective Shigella vaccine based on the GMMA-technology.

## Author Contributions

**Conceptualization:** Francesco Citiulo, Rino Rappuoli, Allan Saul, Laura B. Martin.

**Data curation:** Francesco Citiulo, Francesca Necchi, Omar Rossi, Maria Grazia Aruta, Renzo Alfini, Simona Rondini, Allan Saul, Laura B. Martin.

**Formal analysis:** Francesco Citiulo, Allan Saul, Laura B. Martin.

**Investigation:** Francesco Citiulo, Francesca Necchi, Francesca Mancini, Omar Rossi, Maria Grazia Aruta, Gianmarco Gasperini, Renzo Alfini, Francesca Micoli.

**Methodology:** Francesco Citiulo, Francesca Necchi, Francesca Mancini, Maria Grazia Aruta, Gianmarco Gasperini, Simona Rondini.

**Project administration:** Laura B. Martin.

**Resources:** Allan Saul, Laura B. Martin.

**Supervision:** Francesca Necchi, Simona Rondini, Francesca Micoli, Laura B. Martin.

**Validation:** Francesca Mancini, Omar Rossi.

**Writing – original draft:** Francesco Citiulo, Rino Rappuoli, Allan Saul, Laura B. Martin.

**Writing – review & editing:** Francesco Citiulo, Francesca Necchi, Francesca Mancini, Omar Rossi, Maria Grazia Aruta, Gianmarco Gasperini, Renzo Alfini, Simona Rondini, Francesca Micoli, Rino Rappuoli, Allan Saul, Laura B. Martin.

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
