## [Decision Letter · Decision Letter 0]

8 Jul 2021

Dear Dr Martin,

Thank you very much for submitting your manuscript "Rationalizing the design of a broad coverage Shigella vaccine based on evaluation of immunological cross-reactivity among S. flexneri serotypes" for consideration at PLOS Neglected Tropical Diseases. As with all papers reviewed by the journal, your manuscript was reviewed by members of the editorial board and by several independent reviewers. In light of the reviews (below this email), we would like to invite the resubmission of a significantly-revised version that takes into account the reviewers' comments. 

We cannot make any decision about publication until we have seen the revised manuscript and your response to the reviewers' comments. Your revised manuscript is also likely to be sent to reviewers for further evaluation.

Sincerely,

Christian E. Demeure, Ph.D.

Guest Editor

Javier Pizarro-Cerda

Deputy Editor

Reviewer's Responses to Questions

**Key Review Criteria Required for Acceptance?**

**Methods**

-Are the objectives of the study clearly articulated with a clear testable hypothesis stated?

-Is the study design appropriate to address the stated objectives?

-Is the population clearly described and appropriate for the hypothesis being tested?

-Is the sample size sufficient to ensure adequate power to address the hypothesis being tested?

-Were correct statistical analysis used to support conclusions?

-Are there concerns about ethical or regulatory requirements being met?

Reviewer #1: (No Response)

Reviewer #2: This is a preclinical study to assess cross reactive immune responses, including functional antibodies, in mice (and rabbits) to a 14 different Shigella GMMA vaccine constructs. 

The heat map methodology is an important element of the paper but is not easy to follow. 

line 201 does this whole explanation relate to the predicted IC50 shown in fig 2C? if so please clarify, and that the section on line 188 relates to the measured SBA in 2B. 

line 203: should this be representative heat map (if referring to methods for fig 1 A), not a theoretical heat map? since theoretical heat map is 1B?

line 206: please clarify how the cut of of 3.9 for heterologous SBA was derived? is this the 'maximum' observed?

Struggled to follow the section on modelled heat map - like 201 onwards. 

The abstracts mentions that the study includes rabbits, but no mention is made here.

Reviewer #3: 1. Line 201- the description of the construction of the modeled SBA heatmap is unclear. 

2. Please comment on the chain length for the OAg produced by the GMMA vaccines. 

3. It appears that different amounts of complement have been used for the measurements of bactericidal activity against various Shigella serotypes (Table S3). This can be a confounder in the interpretation of the results. How were the assays harmonized for valid comparison across strains? 

4. Line 191. Explain why the heat inactivation was done at 54oC for 1h as opposed to the usual 56oC for 30 min.

**Results**

-Does the analysis presented match the analysis plan?

-Are the results clearly and completely presented?

-Are the figures (Tables, Images) of sufficient quality for clarity?

Reviewer #1: (No Response)

Reviewer #2: line 235: threshold of cross reactivity in FACS analysis was assigned based on a historical preclinical challenge study that assessed 2 serotypes: flex 3a and 2a. this is presented as a threshold that is 'predictive of field cross coverage'. suggest to include reference to limitations of this extrapolation, including that it is based on only one heterologous assessment. essentially this is one benchmark.

line 306: 'The breadth of the specificity was a little greater as judged by SBA than FACS' - this seems odd. Could the threshold for coding the heat maps be lacking in sensitivity? 

line 316: selection of 1b and 3a based on cross reactivity data: It is not clear why, on the basis of figures 2A and 2B, these sequences were selected. for example, 1c has very similar values to 1b? 3b actually has higher values than 3a in many instances for FACS, but not so for SBA. please expand basis/rationale for serotype selection

line 319: would be useful to mention why sonnei was included based on the results - obvious, but a statement on this would be helpful nonetheless. 

line 323 mention of rabbit results being supportive but there is not mention of the rabbit study throughout (apart from abstract). suggest to remove.

Reviewer #3: 1. A major concern is that the GMMAs contain molecules other than LPS (e.g. outer membrane proteins) and therefore cross reactivity may not be due to O-antigen alone. Sera directed against GMMA from O-negative strain (Line 279) was used to conclude that the antibodies examined are directed against the LPS only (Figure 2S). However, there is no information to indicate the protein composition of the different GMMAs. According to Table S4, the GMMAs have quite different O/protein ratios. A more convincing demonstration of specificity (for example, complete O antigen inhibition of reactivity) would be needed. 

2. The claim that a vaccine containing such serotypes can be broadly protective needs to be confirmed by protective efficacy in vivo.

3. The supplementary material has important information that could be better appreciated within the main manuscript. A re-evaluation of all figures is in order; see comment below.

4. In Lines 317-319, the authors indicate that humans recognize some Shigella serospecificities differently to mice. In Figure S1, the authors show that sera from rabbits immunized with S. flexneri 2a GMMA elicited SBA against S. flexneri 6, which is similar to what has been observed in humans. Since the rabbit data may more closely resemble human data, Figure S1 seems more relevant than Figure 2. 

5. It is not clear the message the authors are trying to convey with the predictions in Figure 2C. Please explain the value of this prediction.

6. Line 235- The authors cite two studies (not mouse or rabbit) when defining the threshold criterion. Please provide more detail on how a threshold criterion was determined, especially without challenge data.

7. Figure S1 (A). Check for consistency in format. Please comment on the comparison of SBA titers if different amount of complement were used.

**Conclusions**

-Are the conclusions supported by the data presented?

-Are the limitations of analysis clearly described?

-Do the authors discuss how these data can be helpful to advance our understanding of the topic under study?

-Is public health relevance addressed?

Reviewer #1: (No Response)

Reviewer #2: The paper concludes that the combination of 4 serotypes identified will provide broad cross reactivity across circulating Shigella, but doesn't contextualise this within the other Shigella serotypes beyond those assessed in this paper. Could some comment be made as to the clinical/epidemiological relevance of the other Shigella serotypes and the expected impact from this combination?

Mention whether preclinical challenge studies were contemplated to confirm in vitro analysis - if not, why not?

other Shigella combo approaches have selected serotype 6, rather than 1b. could the authors comment on the relevance/implications of this, based on their analysis? 

would be helpful for the authors to mention the limitations/assumptions underpinning this analysis and their conclusions

line 408: would be helpful to outline the cross over studies that should be done with human studies where sera are available

Reviewer #3: 1. While the predicted SBA heatmap (Figure 2C) shows that Noriega et al’s bivalent 2a and 3a and the vaccine proposed by Livio et al (S. flexneri 2a, 3a, 6 and S. sonnei) would not protect, Figure 2B predicts that both would be protective, and even more so, Figure S1. Please address this observation in the discussion. 

2. While the inclusion of the S. sonnei GMMA is helpful as a proof-of-concept, it is not clear why it was included in the vaccine, as the manuscript is mainly about cross-protection of S. flexneri.

3. It is unexpected that the bactericidal activity was not the highest for the homologous sera and serotype. Higher titers were obtained in a cross reactive manner. Again, can this be an artifact of the experimental assay conditions? This issue should be addressed/discussed.

4. In vivo protective data would be important to support the broad coverage conclusions.

**Editorial and Data Presentation Modifications?**

Reviewer #1: (No Response)

Reviewer #2: Line 18: mentions rabbits, but not included in description of study in line9, or the rest of the paper. suggest to reconcile/remove

line 19: add...'evaluated in this study' after flexneri serotypes

line 25: suggest to add 'cost' to complexity

line 30: basis, not bases

line 53: suggest to change this to 'These data highlight the difficulties... if based solely on the basis of epidemiology' - as more relevant to the narrative that proceeds it. The difficulty of selection based on subtype/serotype specificity could be positioned around line 82.

line 56: change to homologous AND cross reactive serotypes

line 57: remove 'whereas'

line 75: remove 'have been'

line 77: Suggest to include This 'selection of sub-types on the basis on group specificity' was the basis for studies....

line 92: suggest change 'the same' to homologous and heterologous

line 291: enter 'than' the minimum criterion...

line 347: suggest change instead for conversely

The figures n the results are loaded back to front; 1 is 2 and vice versa

Reviewer #3: 1. Figures 1 and 2 are inverted. Images of higher resolution should be provided. Also, the figure legends should be self explanatory (i.e. shading explained). The Figures 1 and 2 heatmaps are not easy to interpret. Using graphs in the Supplementary Figure S1 (with actual titers) is recommended for ease of visualization and interpretation.

2. Table S3 check for inconsistencies in font (X and Y vs. x and y).

**Summary and General Comments**

Reviewer #1: (No Response)

Reviewer #2: This is an important manuscript that seeks to provide the rationale for the selection of serotypes and subtypes for inclusion in a broadly cross reactive Shigella vaccine. The paper reads well but could benefit from improved clarify in some areas, as identified above. 

The abstract mentions that the study extended to assess the immunogenicity in rabbits but no mention is made in the rest of the paper, except to support conclusions. Suggest to expand or remove.

Reviewer #3: This manuscript describes the capacity of mouse antibodies elicited by vaccination with outer membrane vesicles (exosomes known as GMMA) from Shigella sonnei, to bind S. sonnei and a panel of 11 different S. flexneri subtypes. GMMAs contain all the outer membrane components of the bacteria from which they are derived and may be engineered to express a modified O antigen. Antibody reactivity was determined in vitro via FACS and bactericidal activity. The broader objective of this work is to generate information that would allow to design a vaccine that provides broad coverage against multiple serotypes. 

Mouse antisera against S. flex serogroup 1 and 3 (specifically serotype 1b and 3a) recognized most of S. flexineri serotypes, indicating a vaccine candidate that contains these two serotypes could yield a broad reactivity/coverage. These results are in line with the vaccine proposed by Livio et al., with similar number of serotypes (4) and differing by one serotype, replacing S. flexneri 6 with S. flexneri 1b. 

The analysis of Shigella O-antigen cross reactivity is a valid and important question. Please see specific comments under Methods, Results, and Conclusions.

PLOS authors have the option to publish the peer review history of their article (what does this mean?). If published, this will include your full peer review and any attached files.

Reviewer #1: No

Reviewer #2: No

Reviewer #3: No
---

## [Editor Report · Decision Letter 1]

21 Sep 2021

Dear Dr Martin,

We are pleased to inform you that your manuscript 'Rationalizing the design of a broad coverage Shigella vaccine based on evaluation of immunological cross-reactivity among S. flexneri serotypes' has been provisionally accepted for publication in PLOS Neglected Tropical Diseases.

Best regards,

Christian E. Demeure, Ph.D.

Guest Editor

Javier Pizarro-Cerda

Deputy Editor

The authors have performed a strong work to answer the reviewers requests, and the paper is now much clearer and easier to understand. It is now ready for publication.

---

## [Editor Report · Acceptance letter]

7 Oct 2021

Dear Dr Martin,

We are delighted to inform you that your manuscript, "Rationalizing the design of a broad coverage Shigella vaccine based on evaluation of immunological cross-reactivity among S. flexneri serotypes," has been formally accepted for publication in PLOS Neglected Tropical Diseases.

Best regards,

Shaden Kamhawi

co-Editor-in-Chief

Paul Brindley

co-Editor-in-Chief
